# Maternal Early-Life Risk Factors and Later Gestational Diabetes Mellitus: A Cross-Sectional Analysis of the UAE Healthy Future Study (UAEHFS)

**DOI:** 10.3390/ijerph191610339

**Published:** 2022-08-19

**Authors:** Nirmin F. Juber, Abdishakur Abdulle, Abdulla AlJunaibi, Abdulla AlNaeemi, Amar Ahmad, Andrea Leinberger-Jabari, Ayesha S. Al Dhaheri, Eiman AlZaabi, Fatima Mezhal, Fatma Al-Maskari, Fatme AlAnouti, Habiba Alsafar, Juma Alkaabi, Laila Abdel Wareth, Mai Aljaber, Marina Kazim, Michael Weitzman, Mohammad Al-Houqani, Mohammed Hag Ali, Naima Oumeziane, Omar El-Shahawy, Scott Sherman, Sharifa AlBlooshi, Syed M. Shah, Tom Loney, Wael Almahmeed, Youssef Idaghdour, Raghib Ali

**Affiliations:** 1Public Health Research Center, New York University Abu Dhabi, Abu Dhabi P.O. Box 129188, United Arab Emirates; 2Department of Pediatrics, Zayed Military Hospital, Abu Dhabi P.O. Box 72763, United Arab Emirates; 3Department of Cardiology, Zayed Military Hospital, Abu Dhabi P.O. Box 72763, United Arab Emirates; 4Department of Nutrition and Health, College of Medicine and Health Sciences, UAE University, Al-Ain P.O. Box 15551, United Arab Emirates; 5Department of Pathology, Sheikh Shakhbout Medical City, Abu Dhabi P.O. Box 11001, United Arab Emirates; 6Institute of Public Health, College of Medicine and Health Sciences, UAE University, Al-Ain P.O. Box 15551, United Arab Emirates; 7Zayed Center for Health Sciences, UAE University, Al-Ain P.O. Box 15551, United Arab Emirates; 8College of Natural and Health Sciences, Zayed University, Abu Dhabi P.O. Box 144534, United Arab Emirates; 9Center for Biotechnology, Khalifa University of Science and Technology, Abu Dhabi P.O. Box 127788, United Arab Emirates; 10Department of Genetics and Molecular Biology, Khalifa University of Science and Technology, Abu Dhabi P.O. Box 127788, United Arab Emirates; 11Department of Biomedical Engineering, Khalifa University of Science and Technology, Abu Dhabi P.O. Box 127788, United Arab Emirates; 12Department of Internal Medicine, College of Medicine and Health Sciences, UAE University, Al-Ain P.O. Box 15551, United Arab Emirates; 13The National Reference Laboratory, Abu Dhabi P.O. Box 92323, United Arab Emirates; 14Healthpoint Hospital, Abu Dhabi P.O. Box 112308, United Arab Emirates; 15Abu Dhabi Blood Bank Services, SEHA, Abu Dhabi P.O. Box 109090, United Arab Emirates; 16Department of Environmental Medicine, New York University of Medicine, New York, NY 10016, USA; 17Department of Medicine, College of Medicine and Health Sciences, UAE University, Al-Ain P.O. Box 15551, United Arab Emirates; 18Faculty of Health Sciences, Higher Colleges of Technology, Abu Dhabi P.O. Box 25026, United Arab Emirates; 19Department of Population Health, New York University School of Medicine, New York, NY 10016, USA; 20College of Natural and Health Sciences, Zayed University, Dubai P.O. Box 19282, United Arab Emirates; 21College of Medicine, Mohammed Bin Rashid University of Medicine and Health Sciences, Dubai P.O. Box 505055, United Arab Emirates; 22Heart and Vascular Institute, Cleveland Clinic Abu Dhabi, Abu Dhabi P.O. Box 112412, United Arab Emirates; 23MRC Epidemiology Unit, University of Cambridge, Cambridge CB2 1TN, UK

**Keywords:** gestational diabetes mellitus, pregnancy, maternal early-life factor, epidemiology, United Arab Emirates, UAE healthy future study, GDM, UAE, UAEHFS

## Abstract

Limited studies have focused on maternal early-life risk factors and the later development of gestational diabetes mellitus (GDM). We aimed to estimate the GDM prevalence and examine the associations of maternal early-life risk factors, namely: maternal birthweight, parental smoking at birth, childhood urbanicity, ever-breastfed, parental education attainment, parental history of diabetes, childhood overall health, childhood body size, and childhood height, with later GDM. This was a retrospective cross-sectional study using the UAE Healthy Future Study (UAEHFS) baseline data (February 2016 to April 2022) on 702 ever-married women aged 18 to 67 years. We fitted a Poisson regression to estimate the risk ratio (RR) for later GDM and its 95% confidence interval (CI). The GDM prevalence was 5.1%. In the fully adjusted model, females with low birthweight were four times more likely (RR 4.04, 95% CI 1.36–12.0) and females with a parental history of diabetes were nearly three times more likely (RR 2.86, 95% CI 1.10–7.43) to report later GDM. In conclusion, maternal birthweight and parental history of diabetes were significantly associated with later GDM. Close glucose monitoring during pregnancy among females with either a low birth weight and/or parental history of diabetes might help to prevent GDM among this high-risk group.

## 1. Introduction

Gestational diabetes mellitus (GDM) is a common and potentially serious pregnancy complication if left untreated that affects one in six pregnancies worldwide [1]. The American Diabetes Association (ADA) defines GDM as glucose intolerance resulting in hyperglycemia first diagnosed in the second or third trimester of pregnancy and it does not exclude the possibility of unrecognized or undiagnosed glucose intolerance in the pre-pregnancy period [2]. Data indicates that the burden of GDM has increased by 10–100% in various populations between 1987 to 2007, increasing the burden of metabolic complications in both mothers and offspring [3]. The median estimates of GDM range from 6 to 13% worldwide with the highest prevalence observed in the Middle East and North Africa (MENA) region [4]. In the MENA region, the GDM prevalence in the United Arab Emirates (UAE) was observed to be the third-highest (13.4%) [5].

GDM is a multifactorial disease in which genetic, epigenetic, and environmental factors contribute to its development [1]. GDM is also considered to be an acute condition with short-term effects for the mother and the offspring since glucose intolerance usually reverts back to normal levels after the pregnancy [6]. However, women with GDM history have an increased risk for diabetes occurrence, compared to those without GDM history [1]. Major risk factors for GDM include being overweight/obese, gaining excessive weight during pregnancy, consuming a western diet, belonging to certain ethnic groups (i.e., Middle Easterners), having advanced maternal age, having low/high maternal birthweight, and having a family history with diabetes [1,7]. Maternal metabolic disturbance, such as GDM, has been known to lead to subsequent fetal metabolic programming, hence, increasing the risk of cardiometabolic disorders in the offspring, including future GDM (transgenerational cycle) [8].

Previous studies have focused on the maternal later-life risk factors to tackle short-term pregnancy and delivery complications due to GDM; however, a shift to focus on earlier prevention is highly needed [6]. There are limited studies that have focused on maternal early-life risk factors and later GDM. To date, previous studies on maternal early-life risk factors and later GDM only focused on maternal birthweight [9,10], parental smoking [11], and parental history of diabetes [7]. To our knowledge, only one recent study has focused on childhood body height and later GDM [9], and two studies on childhood body mass index (BMI) and later GDM [10,12]. In addition, other domains of maternal early-life risk factors such as ever-breastfed history, overall health up to 10 years old, and urbanicity at birth have not been explored.

Our analysis of data from the UAE Healthy Future Study (UAEHFS) aimed to be the first population-based study characterizing the associations between several maternal early-life factors and later GDM in a single study. Our study aimed to estimate the GDM prevalence and to comprehensively examine the associations of several maternal early-life risk factors and later GDM, including commonly studied (maternal birthweight, parental smoking, and parental history of diabetes), less studied (parental education attainment, childhood body size, and height), and unstudied maternal early-life risk factors (urbanicity at birth, ever-breastfed, and childhood overall health), with later GDM.

## 2. Materials and Methods

### 2.1. Study Design, Participants, and Setting

This is a retrospective cross-sectional study using the UAE Healthy Future Study (UAEHFS) data collected from February 2016 to April 2022. We studied 702 ever-married women aged 18–67 years who had complete information on diabetes diagnosis history during their lifetime (Figure 1). The study design, questionnaire, and methodologies of the UAEHFS are described elsewhere [13]. In brief, the UAEHFS is a longitudinal population-based cohort study that aims to explore risk factors for non-communicable diseases (NCDs) among Emirati nationals aged 18 years or above. Emirati adults were asked to fill out the questionnaire and had some physical measurements taken at multiple centers across major cities in the UAE: Abu Dhabi, Al-Ain, Dubai, and Ras Al Khaimah. Due to the COVID-19 pandemic, study recruitment shifted to online enrollment starting from April 2020, and an online questionnaire was introduced to the new participants. Physical measurements, such as BMI, were taken in the participating centers for new participants that filled and returned the online questionnaire once in-person recruitment resumed in September 2020.

### 2.2. Measurements

#### 2.2.1. Outcome Measure

We analyzed self-reported physician diagnoses for diabetes based on the questionnaire response to: “Has a doctor ever told you that you have diabetes?” (yes, no), and GDM as an outcome variable was extracted from the response to the following questionnaire item: “Did you only have diabetes during pregnancy?” (yes, no). GDM in this study was defined as females diagnosed with diabetes during pregnancy only and we excluded those with non-GDM diabetes [14].

#### 2.2.2. Maternal Early-Life Risk Factors

We considered nine variables as predictors of interest: maternal birthweight, parental smoking at birth, urbanicity at birth, ever-breastfed history, parental education attainment, parental history of diabetes, overall health up to 10 years old, as well as body size and height at 10 years old. Maternal birthweight was determined based on the response to questionnaire item: “What was your birth weight (in kg)?”, and we categorized maternal birthweight into low maternal birthweight (<2.5 kg) and normal maternal birthweight (≥2.5 kg) [15]. Less than 1% of our study participants reported a maternal birthweight above 4 kg (high maternal birthweight), therefore we labeled the 2.5 kg and above as normal maternal birthweight category. Parental smoking (at birth) was constructed based on the questionnaire response to “Did your mother or father smoke regularly around the time when you were born?” (yes, no). Urbanicity at birth was determined based on the questionnaire response to: “Where do you and your family live around the time of your birth?”. We categorized city as an urban area and non-cities (village, desert, island, and others) as rural or other non-urban. Ever-breastfed history was determined based on the questionnaire response to: “Were you breastfed when you were a baby?” (yes, no). Parental education attainment was constructed based on the questionnaire response to: “What level of education did your father or mother complete?”, and we categorized parental education, based on the maximum level of either father or mother education reported, into 6 years or below, >6–12 years, and >12 years of schooling. We also studied maternal early-life factors during childhood which included overall health up to 10 years old, as well as body size and height at 10 years old. Overall health up 10 years old was determined based on the questionnaire response to: “In general, how was your health in childhood (less than 10 years old)?” (poor or fair, good or excellent). Body size and height at 10 years old were determined based on the respective questionnaire responses to: “When you were 10 years old, compared to average would you describe yourself as: about average, thinner, or plumper?”, and “When you were 10 years old, compared to average would you describe yourself as: about average, shorter, or taller?”. Lastly, parental history of diabetes was constructed based on the questionnaire response to” “Did your mother or father ever suffer from diabetes?” (yes, no). Due to a significant number of uncertain values (missing or prefer not to answer (PNA) or do not know (DN)), we used missing indicator by creating “Missing/PNA/DN” variable for each variable that contains uncertain values.

#### 2.2.3. Sociodemographic Factors, Health Factors, and BMI

Age was constructed based on the questionnaire response to “What is your date of birth”. Urbanicity (current) was determined based on the questionnaire response to: “Where do you and your family live now?”. We categorized city as an urban area and non-cities (village, dessert, island, and others) as rural or other non-urban. Marital status was determined based on the questionnaire response to: “What is your marital status?” (married, divorced or separated, widow or widower). Education attainment was determined based on the response to the questionnaire item: “What is the highest level of education that you have completed?”. We then categorized education levels into three categories (6 years or below, >6 to 12 years, and >12 years). Health factors such as current overall health and history of poly-cystic ovarian syndrome were determined based on the questionnaire responses to: “In general, how would you rate your overall health now?” (poor or fair, good or excellent) and “Has a doctor ever told you that you have polycystic ovarian syndrome/disease” (yes, no). Lastly, BMI was calculated using the Tanita MC 780 (Tanita Inc., Tokyo, Japan) by nurses at the recruitment centers [12]. We also categorized BMI into three categories: normal or below (<25 kg/m^2^), overweight (25–<30 kg/m^2^), and obese (≥30 kg/m^2^) [16].

### 2.3. Statistical Analysis

Current and early-life characteristics of the study participants were evaluated using frequencies with percentage (*n*, %) for categorical variables and means with standard deviations (means ± SD) for continuous variables (Table 1 and Table 2). We used chi-squared tests for categorical variables and *t*-tests for continuous variables to compare distributions of study participants based on their GDM status (without vs. with GDM). For each maternal early-life risk factor included in this study, we fitted a Poisson regression model with robust variance to estimate the risk ratio (RR) and its 95% confidence interval (95% CI) to assess its association with later GDM [17]. We examined the RR and its 95% CI under three models: unadjusted (crude), age-and-parental history of diabetes (basic model), and full adjustment (fully adjusted) models. We adjusted for age-and parental history of diabetes since these factors have been known to strongly confound the associations involving GDM as an outcome [7]. We further added relevant potential confounding factors in the full adjustment model for each maternal early-life risk factor to better estimate the respective association between maternal early-life risk factor and later GDM (i.e., urbanicity at birth was adjusted for parental education, parental smoking, and birth weight). The complete list of relevant confounding factors is shown in Table 3. The missing indicator method was used to handle uncertain values from missing, prefer not to answer (PNA), and do not know (DN) responses. In addition, we also performed a sensitivity analysis to include only those with uncertain values of diabetes history to examine the pattern of missingness (Table 1). Analyses were carried out using STATA 17.0 (StataCorp, Brazos, TX, USA). *p* values < 0.05 were considered statistically significant.

### 2.4. Ethical Approval

The study and its procedures have been reviewed and approved by the Institutional Review Board at New York University Abu Dhabi, Dubai Health Authority, Ministry of Health and Prevention in the UAE, and Health Research and Technology Committee, reference number DOH/HQD/2020/516. Written consent was obtained from participants at the centers or by filling out an online consent form before data collection started.

## 3. Results

Table 1 summarizes the current characteristics of the study participants at baseline based on GDM status. Out of 702 ever-married females in our study, 36 females reported ever being diagnosed with diabetes only during pregnancy (GDM prevalence = 5.1%). Compared to those without GDM, females with GDM were older (36.5 ± 8.5 vs. 33.0 ± 7.5 years), had higher BMI (31.2 ± 6.0 vs. 27.5 ± 5.8 kg/m^2^), a greater proportion were classified as obese (47.2% vs. 24.0), had lower levels of education (50.0% vs. 61.7% > 12 years of schooling), and a higher proportion being in poor or fair health (41.7% vs. 22.8%).

Table 2 shows the early-life characteristics of the study participants based on GDM status. Compared to those without GDM, a higher proportion of females with GDM reported having low maternal birthweight of <2.5 kg (61.1% vs. 54.2%) and having a parental history of diabetes (86.1% vs. 64.3%). Our sensitivity analysis (Table 1 and Table 2) revealed non-systematic differences in missingness among those excluded due to missing values of diabetes status.

Table 3 presents the risk ratios (RRs) of the associations between maternal early-life risk factors and later GDM. Under three models, compared to those with normal maternal birthweight (≥2.5 kg), those with low maternal birthweight (<2.5 kg) were four times more likely to report GDM (RR = 3.67, 95% CI: 1.26–10.7 in the crude model; RR = 4.13, 95% CI: 1.42–12.1 in the basic model; RR = 4.04, 95% CI: 1.36–12.0 in fully adjusted model). Similarly, those with parental history of diabetes were three times more likely to report GDM in all models (RR = 3.28, 95% CI: 1.29–8.34 in the crude model, and RR = 2.86, 95% CI: 1.10–7.43 in basic or fully adjusted model), compared to those without a parental history of diabetes. Other domains of maternal early-life risk factors were shown to be not statistically significant in all models, however, parental smoking at birth as well as body size and height at 10 years old were shown to have a strong magnitude of association with later GDM. In the fully adjusted model, compared to those without parental smoking at birth, females with parental smoking at birth had a 24% increased risk of later GDM. Females with plumper body size and shorter height at 10 years old were associated with 38% and 37% increased risk of later GDM, compared to those with average body size and height at 10 years old, respectively. Interestingly, those who reported taller height at 10 years old were associated with a 56% decreased risk of later GDM, compared to females that reported average height at 10 years old.

Figure 2 illustrates the prevalence of GDM by each maternal early-life factor in our study. GDM prevalence for each variables of maternal early-life risk factors as follows. Maternal birthweight: <2.5 kg (8.6%), ≥2.5 kg (2.3%), and missing/DN/PNA (5.7%). Parental smoking at birth: no (4.7%), yes (6.1), and missing/DN/PNA (6.7%). Parental history of diabetes: no (2.1%), and yes (6.8%). Parental education: ≤12 years (7.3%), >12 years (7.3%), and missing/DN/PNA (3.6%). Body size at 10 years old: thinner (5.2%), about average (4.7%), plumper (6.1%), and missing/DN/PNA (5.4%). Body height at 10 years old: shorter (6.8%), about average (5.1%), taller (2.2%), and missing/DN/PNA (8.5%). Urbanicity at birth: rural or other non-urban (4.2%), urban (4.9%), and missing/DN/PNA (6.4%). Ever-breastfed: no (5.6%), yes (5.0%), and missing/DN/PNA (5.4%). Overall health up to 10 years old: poor or fair (7.8%), and good or excellent (4.9%).

## 4. Discussion

In this retrospective cross-sectional study, we found that 5.1% of ever-married women reported ever being diagnosed with diabetes only during pregnancy (GDM). The prevalence of GDM by each maternal early-life factor in our study ranged from 2.1% to 8.6% (Figure 2). The GDM prevalence in our study was shown to be lower than national estimates across the MENA region and in the UAE, which revealed approximately 13% of GDM prevalence [4,5]. Previous studies have found that Asian women were among the high-risk group for developing GDM [31,32]. Our study included females aged 18 to 67 years, and a previous study has shown that GDM prevalence in women aged ≥ 30 years was 2.26 times higher compared to women aged 15–29 years [5]. Advanced maternal age is known to be a significant risk factor for GDM, as the risk of developing GDM was 7 to 10 times higher in pregnant women aged 24 years and above compared to their younger counterparts [33]. Furthermore, outcome ascertainment utilized self-reported and may have also contributed to the lower prevalence estimate in our study. For comparison, two Omani studies using self-reported and medical records data revealed the GDM prevalence of 3.3% and 7.6%, respectively [34,35].

We found significant associations between low maternal birthweight (<2.5 kg) and later GDM in the basic model and fully adjusted model, compared to females with normal maternal birthweight history (≥2.5 kg). Our results are consistent with previous studies that have identified the association between low maternal birthweight and later GDM [1,36]. Previous studies also found “U-shaped” associations between maternal birthweight and later GDM, in which high maternal birthweight (>4 kg) as well as low maternal birthweight (<2.0 kg or <2.5 kg,) were shown to be significantly associated with GDM, compared to normal maternal birthweight [9]. In our study, less than one percent reported maternal birthweight above 4 kg, therefore we could not analyze the association between high maternal birthweight and later GDM, and we included those with >4 kg of maternal birthweight in the normal maternal birthweight category (≥2.5 kg). Another study that was able to evaluate low and high maternal birthweight only found a significant association between low birthweight and later GDM [37]. Low maternal birthweight and GDM are linked due to insulin resistance, as low maternal birthweight reflects undernutrition in the womb and therefore alters the expression of genes, hence leading to future metabolic consequences including GDM [1].

Our study also found significant associations between parental history of diabetes and later GDM in all models. Our results are in general agreement with previous studies that have revealed similar associations with GDM [7,38,39,40]. One meta-analysis study pooled 33 studies on family history with diabetes and GDM showed an overall odds ratio of 3.46 for the association between family history with diabetes and later GDM [40], similar to our study. To date, the level of evidence on the association between parental/family history with diabetes and later GDM is shown to be highly suggestive [38], and the link between parental/family history with diabetes and later GDM lies in genetic and lifestyle factors with metabolic programming due to family/parental history of diabetes proposed as one mechanism [8,41]. Furthermore, a previous study has shown an intergenerational cycle of GDM in which females with GDM mothers are more likely to experience GDM in their pregnancies [42]. Family history of diabetes is an important risk factor for later GDM, and a previous study revealed that 40% of women with GDM were more likely to have had a first-degree relative with diabetes [43].

We found no statistically significant associations between parental smoking at birth and later GDM in all models. A previous study has shown that paternal smoking during pregnancy was not associated with the risk of GDM in the daughter and maternal heavy smoking was shown to be significantly associated with later GDM [10]. A previous study from the UAE Health Future (UAEHFS) Pilot Study that used biochemical verification for smoking status revealed that 42% of males and 9% of females had recent tobacco use [44]. We believe that the low prevalence of maternal smoking in our study population contributed to non-significant associations between parental smoking at birth and later GDM.

We found non-significant associations between parental education and later GDM; however, the magnitude of associations between high parental education and later GDM was shown to have an 8–9% increase for GDM risk in the basic and fully adjusted models, compared to those with the lowest parental education level (6 years or below of schooling). In our study, parental education contained many missing values (>50%), therefore, a small sample size, especially with four categorizations of parental education that we used, led to low statistical power to detect differences. We have similar findings to a recent study that revealed a non-significant association between maternal education attainment and GDM risk [45]. Parental education attainment has been known to be a proxy for family background [46], but the role of parental education and GDM risk is not fully known.

Our study also examined other domains of early maternal life-risk factors that were less commonly studied, namely childhood body size and height at 10 years old. We did not find any significant association in the aforementioned domains of maternal early-life factors due to a low catchment of those with GDM in our study, hence low statistical power to detect differences. Regarding the magnitude of associations, we found those who reported having thinner or plumper body size at 10 years old to be positively associated with later GDM, compared to those with average body size. While two recent studies have confirmed a positive association between childhood or adolescent obesity and later GDM [10,12], the “U-shaped” relationship between childhood body size and later GDM in our study provided new information and is worthy of further investigation. We also found that compared to those with average body height at 10 years old, those who reported having a shorter height at 10 years old were positively associated with later GDM. Conversely, those reported having a taller height at 10 years old were negatively associated with later GDM. One recent study has confirmed the association between shorter childhood height and later GDM [10]; however, more studies are needed to confirm the association between taller childhood height and later GDM. Childhood body size and height are proxies for fetal growth and nutrition and are shown to be important determinants not just for childhood health but also for future health in adulthood [47], so more studies are needed to elucidate the exact mechanism of childhood body size and height with GDM risk.

In this study, we proposed novel maternal early-life risk factors namely: ever-breastfed history, overall health up to 10 years old, and urbanicity at birth. Despite non-significant relationships in all models for these maternal early-life risk factors, the magnitudes of associations from these domains were shown to be worth further investigation. We found that those with ever-breastfed history had a weak negative association with later GDM, compared to those with never-breastfed history. We proposed a new direction of the association between ever-breastfed and later GDM. A previous study found that mothers with GDM had a shorter breastfeeding duration compared to mothers without GDM, but breastfeeding was found to be beneficial for mothers with GDM as their lipid and glucose metabolic profiles improved [48]. We also found that those who reported having good or excellent health status up to 10 years old were negatively associated with later GDM, compared to those with poor or fair health. Overall health status has been known as one of the comorbidity indices commonly used in health research [49]. Lastly, we found that those born in urban areas had a weak positive association with later GDM, compared to those in rural or other non-urban regions. A previous study found an increased odds of GDM among women living in urban areas compared to those in rural areas [50], and proposed urban lifestyle factors (western diet, low physical activity, and biochemical risk factors) as possible mechanisms [51].

### Strengths and Limitations

To our knowledge, this is the first comprehensive population-based study examining the associations between several maternal early-life factors and later GDM, including previously unstudied maternal early-life risk factors such as overall health status, body size, and height during childhood. We were able to control for confounding factors for each maternal early-life risk factor that is relevant as stated in the literature. We excluded never married females and were therefore able to address immortal-time bias, as they do not have the same chance to experience GDM based on cultural and religious norms in the UAE country setting [52]. Similarly, to separate the effect of non-pregnancy-related diabetes diagnosis, we excluded those with non-GDM diabetes, since the genetic risk factors and pathogenesis of these two diseases are known to differ [53]. Finally, we tried to maximize the representativeness of the sample by recruiting participants from multiple centers in four major cities in the UAE. Our study was subject to several limitations. One major limitation was self-reported GDM status, raising the concern of disease ascertainment accuracy. A previously validated study has shown self-reported GDM to be accurate with 94% of the self-reported GDM cases being confirmed by a physician [54]. In addition, maternal early-life factors in our study were assessed based on self-reporting, hence, we cannot rule out recall error. However, we adjusted for age in our basic and fully adjusted models to address the possible age effects of recall error. We used ever-married females as a proxy for ever-pregnant females for the GDM outcome in our study. A previous study in the UAE country setting in 2008 revealed that more than 95% of ever-married women of reproductive age (15–49 years of age) reported having one or more children [55]. Furthermore, we had a lot of uncertain or missing values for some predictors; however, our sensitivity analyses revealed non-systematic mechanisms in the missingness pattern. Lastly, our study may be prone to residual confounding factors due to the nature of the observational research.

## 5. Conclusions

Maternal birthweight and parental history of diabetes were significantly associated with later GDM. Close glucose monitoring during pregnancy among females with either a low birth weight and/or parental history of diabetes might help to prevent GDM among this high-risk group.

## Figures and Tables

**Figure 1 ijerph-19-10339-f001:**
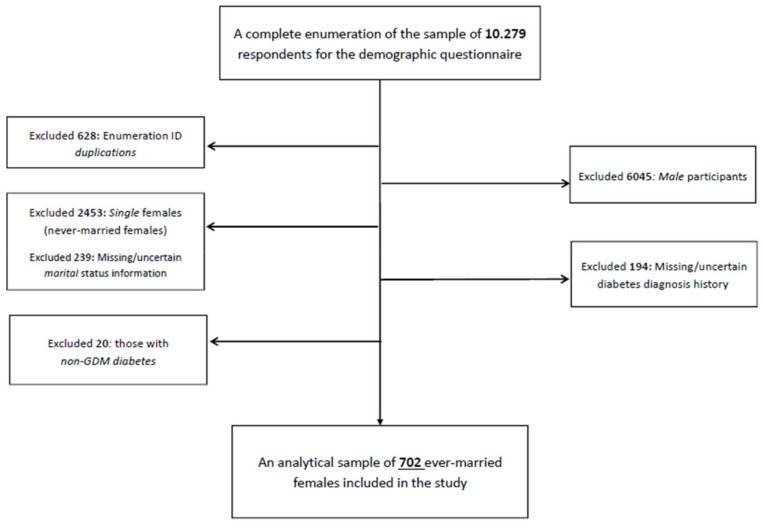
Flowchart of the final analytical sample included in the study.

**Figure 2 ijerph-19-10339-f002:**
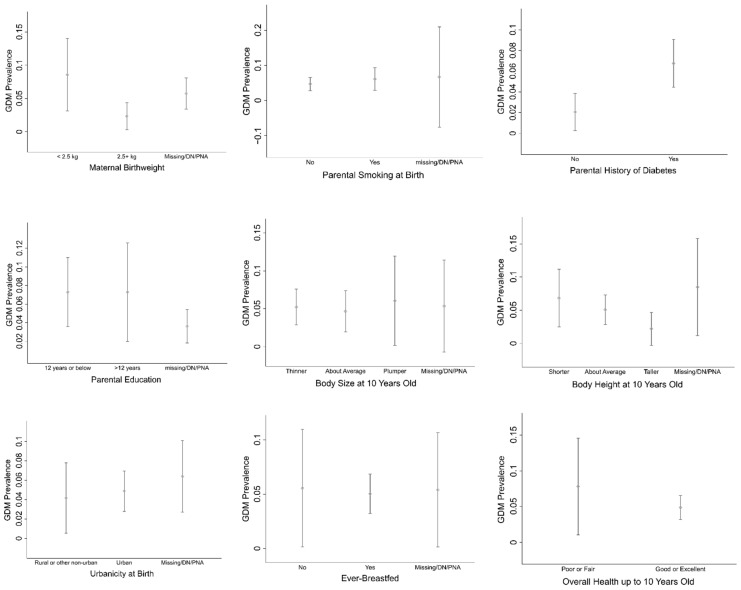
Prevalence of GDM by each maternal early−life risk factor. Point estimates and 95% CI are reported.

**Table 1 ijerph-19-10339-t001:** Current characteristics of the participants at questionnaire based on GDM (N = 702).

Characteristics	Without GDM (N = 666) *	With GDM (N = 36)	Missing/PNA/DN on Diabetes History (N = 194)	*p*-Value *
Age year (mean ± SD)	33.0 ± 7.5	36.5 ± 8.5	36.5 ± 8.3	**0.022**
BMI, year (kg/m^2^ ± SD)	27.5 ± 5.8	31.2 ± 6.0	28.7 ± 6.3	**0.002**
BMI levels, n (%)				**0.008**
Normal or below (<25 kg/m^2^)	201 (30.2)	4 (11.1)	29 (15.0)	
Overweight (25 to <30 kg/m^2^)	194 (29.1)	9 (25.0)	48 (24.7)	
Obese (30+ kg/m^2^)	160 (24.0)	17 (47.2)	43 (22.2)	
Missing	111 (16.7)	6 (16.7)	74 (38.1)	
Marital status, n (%)				0.148
Married	545 (81.8)	34 (94.4)	163 (84.0)	
Divorced or separated	111 (16.7)	2 (5.6)	27 (13.9)	
Widow or widower	10 (1.5)	0 (0)	4 (2.1)	
Urbanicity, n (%)				0.696
Rural or other non–urbans	61 (9.2)	2 (5.6)	9 (4.6)	
Urban (city)	547 (82.1)	30 (83.3)	87 (44.9)	
Missing/PNA/DN	58 (8.7)	4 (11.1)	98 (50.5)	
Education attainment, n (%)				**0.002**
6 years of schooling or below	20 (3.0)	5 (13.9)	67 (34.5)	
>6–12 years of schooling	235 (35.3)	13 (36.1)	46 (23.7)	
>12 years of schooling	411 (61.7)	18 (50.0)	81 (41.8)	
Current overal health, n (%)				**0.010**
Poor or fair	152 (22.8)	15 (41.7)	177 (91.24)	
Good or excellent	514 (77.2)	21 (58.3)	17 (8.76)	
With poly–cystic ovarian syndrome history, n (%)				0.237
No	460 (69.1)	21 (58.3)	6 (3.1)	
Yes	127 (19.1)	11 (30.6)	4 (2.1)	
Missing/PNA/DN	79 (11.9)	4 (11.1)	184 (94.9)	

* Compares without and with GDM groups using the *t*-test for continuous variables and the χ^2^ for categorical variables.

**Table 2 ijerph-19-10339-t002:** Early-life characteristics of the participants based on GDM (N = 702).

Maternal Early-Life Factors	Without GDM (N = 666) *	With GDM (N = 36)	Missing/PNA/DN on Diabetes History (N = 194)	*p*-Value *
Own birthweight, n (%)				**0.043**
2.5+ kg (normal birthweight)	209 (31.4)	5 (13.9)	8 (4.1)	
<2.5 kg (low birthweight)	96 (14.4)	9 (25.0)	1 (0.5)	
Missing/PNA/DN	361 (54.2)	22 (61.1)	185 (95.4)	
Parental smoking at birth, n (%)				0.709
No	451 (67.7)	22 (61.1)	14 (7.2)	
Yes	201 (30.2)	13 (36.1)	6 (3.1)	
Missing/PNA/DN	14 (2.1)	1 (2.8)	174 (89.7)	
Urbanicity at birth, n (%)				0.654
Rural or other non-urbans	115 (17.3)	5 (13.9)	18 (9.3)	
Urban (city)	390 (58.6)	20 (55.6)	59 (30.4)	
Missing/PNA/DN	161 (24.2)	11 (30.6)	117 (60.3)	
Ever-breasfed history, n (%)				0.976
No	68 (10.2)	4 (11.1)	1 (0.5)	
Yes	528 (79.3)	28 (77.8)	19 (9.8)	
Missing/PNA/DN	70 (10.5)	4 (11.1)	174 (89.7)	
Parental education, n (%)				0.163
6 years of schooling or below	88 (13.2)	8 (22.2)	3 (1.6)	
>6–12 years of schooling	90 (13.5)	6 (16.7)	11 (5.7)	
>12 years of schooling	89 (13.4)	7 (19.4)	7 (3.6)	
Missing/PNA/DN	399 (59.9)	15 (41.7)	173 (89.2)	
Overal health up to 10 years old, n (%)				0.307
Poor or fair	59 (8.9)	5 (13.9)	19 (9.8)	
Good or excellent	607 (91.1)	31 (86.1)	1 (0.5)	
Missing/PNA/DN			174 (89.7)	
Body size at 10 years of age, n (%)				0.971
About average	225 (33.8)	11 (30.6)	9 (4.6)	
Thinner	326 (49.0)	18 (50.0)	9 (4.6)	
Plumper	62 (9.3)	4 (11.1)	2 (1.0)	
Missing/PNA/DN	53 (8.0)	3 (8.3)	174 (89.7)	
Body height at 10 years of age, n (%)				0.206
About average	355 (53.3)	19 (52.8)	12 (6.2)	
Shorter	123 (18.5)	9 (25.0)	6 (3.1)	
Taller	134 (20.1)	3 (8.3)	2 (1.0)	
Missing/PNA/DN	54 (8.1)	5 (13.9)	174 (89.7)	
Parental history with diabetes, n (%)				**0.007**
No	238 (35.7)	5 (13.9)		
Yes	428 (64.3)	31 (86.1)	13 (6.7)	
Missing/PNA/DN			181 (93.3)	

* Compares without and with GDM groups using the *t*-test for continuous variables and the χ^2^ for categorical variables.

**Table 3 ijerph-19-10339-t003:** Associations between maternal early-life risk factors and later GDM among 702 ever-married females.

Early Life Risk Factors	Crude Model		Age and Parental Diabetes Adjusted Model		Confounding Factors Adjusted Model ^a^		Confounding Factors
	RR [95% CI]	*p*-Value	RR [95% CI]	*p*-Value	RR [95% CI]	*p*-Value	
Maternal birthweight							Parental education, parental smoking [9,15].
2.5+ kg (normal birthweight)	(Reference)		(Reference)		(Reference)		
<2.5 kg (low birthweight)	**3.67 [1.26–10.7]**	**0.017**	**4.13 [1.42–12.1]**	**0.009**	**4.04 [1.36–12.0]**	**0.012**	
Missing/PNA/DN	2.46 [0.94–6.40]	0.065	2.08 [0.80–5.42]	0.132	2.06 [0.80–5.33]	0.137	
Parental smoking at birth, n (%)							Parental education, birthweight, ever-breastfed history [9,18].
No	(Reference)		(Reference)		(Reference)		
Yes	1.31 [0.67–2.54]	0.433	1.14 [0.59–2.19]	0.705	1.24 [0.65–2.40]	0.508	
Missing/PNA/DN	1.43 [0.21–9.96]	0.716	1.43 [0.19–10.5]	0.727	1.77 [0.29–11.0]	0.538	
Urbanicity at birth, n (%)							Parental education, parental smoking, birthweight [15,19,20].
Rural or other non-urbans	(Reference)		(Reference)		(Reference)		
Urban (city)	1.17 [0.45–3.06]	0.747	0.98 [0.38–2.54]	0.960	1.03 [0.39–2.67]	0.960	
Missing/PNA/DN	1.54 [0.55–4.31]	0.416	0.83 [0.27–2.51]	0.737	0.65 [0.21–2.04]	0.462	
Ever-breastfed history, n (%)							Parental education, parental smoking [18,21].
No	(Reference)		(Reference)		(Reference)		
Yes	0.91 [0.33–2.51]	0.850	0.91 [0.33–2.50]	0.857	0.95 [0.35–2.58]	0.915	
Missing/PNA/DN	0.97 [0.25–3.75]	0.968	0.96 [0.25–3.62]	0.950	0.99 [0.27–3.70]	0.992	
Parental education, n (%)							Parental smoking [22].
6 years of schooling or below	(Reference)		(Reference)		(Reference)		
>6–12 years of schooling	0.75 [0.27–2.08]	0.581	0.86 [0.31–2.41]	0.777	0.85 [0.30–2.37]	0.753	
>12 years of schooling	0.88 [0.33–2.32]	0.788	1.09 [0.41–2.89]	0.859	1.08 [0.41–2.84]	0.884	
Missing/PNA/DN	**0.44 [0.19–0.99]**	**0.049**	0.59 [0.25–1.40]	0.229	0.56 [0.23–1.34]	0.193	
Overall health up to 10 years old, n (%)							Parental education, parental smoking, birthweight, body size at 10 years old [15,23,24].
Poor or fair	(Reference)		(Reference)		(Reference)		
Good or excellent	0.62 [0.25–1.55]	0.306	0.64 [0.26–1.56]	0.326	0.68 [0.28–1.61]	0.374	
Body size at 10 years of age, n (%)							
About average	(Reference)		(Reference)		(Reference)		Parental education, parental smoking, birthweight, all health at 10 years old [10,24,25,26,27].
Thinner	1.12 [0.54–2.33]	0.757	1.02 [0.50–2.10]	0.950	1.05 [0.51–2.19]	0.894	
Plumper	1.30 [0.43–3.95]	0.644	1.23 [0.40–3.78]	0.724	1.38 [0.46–4.16]	0.564	
Missing/PNA/DN	1.15 [0.33–3.99]	0.826	0.93 [0.26–3.28]	0.906	1.01 [0.25–4.11]	0.984	
Body height at 10 years of age, n (%)							
About average	(Reference)		(Reference)		(Reference)		Birthweight, overall health at 10 years old [28,29,30].
Shorter	1.34 [6.22–2.89]	0.453	1.43 [0.67–3.07]	0.358	1.37 [0.64–2.91]	0.416	
Taller	0.43 [0.13–1.44]	0.170	0.42 [0.13–1.40]	0.159	0.44 [0.14–1.44]	0.176	
Missing/PNA/DN	1.67 [0.65–4.30]	0.289	1.66 [0.64–4.26]	0.296	1.54 [0.57–4.11]	0.393	
Parental history with diabetes, n (%)							None
No	(Reference)		(Reference)				
Yes	**3.28 [1.29–8.34]**	**0.012**	**2.86 [1.10–7.43]**	**0.031**			

RR: risk ratio; CI: confidence interval; PNA: prefer not to answer; DN: do not know. ^a^ Basic model plus additional adjustments.

## Data Availability

The datasets used and/or analyzed during the current study are available from the senior author on reasonable request.

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
