# Peer review of "Maternal Early-Life Risk Factors and Later Gestational Diabetes Mellitus: A Cross-Sectional Analysis of the UAE Healthy Future Study (UAEHFS)"

_ijerph, 2022, doi:10.3390/ijerph191610339_

Round 1
Reviewer 1 Report
This is well written paper and its originality is high. Though, there are some points to be revised.
1. 702 participants analyzed in this study were ever-married females. To use RR(risk ratio), 702 participants need to be exposed to a risk factor, which is pregnancy. Therefore, the group should be......1) the ones who were ever-pregnant had experienced GDM 2) the ones who were ever-pregnant had not experienced GDM
2. The authors had examined RR under three models. Unadjusted, basic, fully adjusted model. It is very interesting, but the authors need to write the evidence in introduction why age-parental history need to be adjusted/ and of potential confounding factors need to be adjusted.
3. The tile 'Prevalence of GDM by each maternal early-life risk factors...' need to input Figure 2. And please write contents in the Results part.
4. Discussion L249-L252 No information was presented in the Results part.
Author Response
RESPONSE LETTER
Thank you for reviewing our manuscript ID IJERPH-1812231. Your comments and suggestions helped us improve this manuscript. Please see below for our point-by-point responses to your comments.
COMMENT #1:
702 participants analyzed in this study were ever-married females. To use RR (risk ratio), 702 participants need to be exposed to a risk factor, which is pregnancy. Therefore, the group should be......1) the ones who were ever-pregnant had experienced GDM 2) the ones who were ever-pregnant had not experienced GDM
Response:
Thank you for the comment and the sharp observation. We would like to emphasize the reason why we compared ever-married without GDM vs ever-married without GDM instead of ever-pregnant with GDM vs ever-pregnant without GDM as the ideal research methodology design. Due to data limitations, we used ever-married females as a proxy for ever-pregnant females for the GDM outcome in our study. A previous study in the UAE country setting in 2008 revealed that only 3.7% of ever-married women of reproductive age (15-49 years of age) reported having no children.1 Therefore, we used ever-married females as a proxy for ever-pregnant females in our study. We have now added information regarding this in Strengths and Limitations (Line 388-391).
“We used ever-married females in our study as a proxy for ever-pregnant females for the GDM outcome. A previous study in the UAE country setting in 2008 revealed that more than 95% of ever-married women of reproductive age (15-49 years of age) reported having one or more children1”.
COMMENT #2:
The authors had examined RR under three models. Unadjusted, basic, fully adjusted model. It is very interesting, but the authors need to write the evidence in introduction why age-parental history need to be adjusted/ and of potential confounding factors need to be adjusted.
Response:
Thank you for the suggestion. We have now added these points under Statistical Analysis section (Line 192-198).
“We adjusted for age-and parental history of diabetes since these factors have been known to strongly confound the associations involving GDM as an outcome.2 We further added relevant potential confounding factors in the full adjustment model for each maternal early-life risk factor to better estimate the respective association between maternal early-life risk factor and later GDM (i.e. urbanicity at birth was adjusted for parental education, parental smoking, and birth weight)”.
COMMENT #3:
The tile 'Prevalence of GDM by each maternal early-life risk factors...' need to input Figure 2. And please write contents in the Results part.
Response:
Thank you for pointing this out. We have now added the title for Figure 2 (Line 266) and added the content in the Result (Line 254-264).
Fig. 2 Prevalence of GDM by each maternal early-life risk factor. Point estimates and 95% CI are reported.
“Figure 2 illustrates the prevalence of GDM by each maternal early-life factor in our study. GDM prevalence for each variables of maternal early-life risk factors as follows. Maternal birthweight: <2.5 kg (8.6%), ≥2.5 kg (2.3%), and missing/DN/PNA (5.7%). Parental smoking at birth: no (4.7%), yes (6.1), and missing/DN/PNA (6.7%). Parental history of diabetes: no (2.1%), and yes (6.8%). Parental education: ≤12 years (7.3%), >12 years (7.3%), and missing/DN/PNA (3.6%). Body size at 10 years old: thinner (5.2%), about average (4.7%), plumper (6.1%), and missing/DN/PNA (5.4%). Body height at 10 years old: shorter (6.8%), about average (5.1%), taller (2.2%), and missing/DN/PNA (8.5%). Urbanicity at birth: rural or other non-urban (4.2%), urban (4.9%), and missing/DN/PNA (6.4%). Ever-breastfed: no (5.6%), yes (5.0%), and missing/DN/PNA (5.4%). Overall health up to 10 years old: poor or fair (7.8%), and good or excellent (4.9%)”.
COMMENT #4:
Discussion L249-L252 No information was presented in the Results part.
Response:
Thank you again and we have now added the content in the Result (Line 254-264).
“As stated in the quotation for comment #3 above”.
- Al Awad M, Chartouni C. Explaining the Decline in Fertility among Citizens of the GCC Countries: the Case of the UAE. Education, Business and Society: Contemporary Middle Eastern Issues. 2014;
- Larrabure-Torrealva GT, Martinez S, Luque-Fernandez MA, et al. Prevalence and risk factors of gestational diabetes mellitus: findings from a universal screening feasibility program in Lima, Peru. BMC pregnancy and childbirth. 2018;18(1):1-9.

Reviewer 2 Report
INTRODUCTION is well written. Although the scientific background for the investigation being taken is well explained, I believe the Introduction could benefit from the bringing in the term Fetal programming since this enthity is responsible for the future perspectives of offspring, especially regarding diabetes mellitus, arterial hypertension and obesity. The introduction of fetal programming in Introduction section is justified by the evaluation of Parental history of diabetes (this could encompass gestational diabetes of the mother), Parental smoking at birth (this could encompass Maternal smoking at birth), etc. MATERIALS AND METHODS are flawlessly written. Population is well defined and sampling approach is clearly written. RESULTS are well presented, without substantional overlapping between data in text and Tables and Figures. However, I believe the addition of p values in Table 1 and Table 2 could increase the quality of presented results. DISCUSSION is well balanced, with carefully observed and presented limmitations and therefore it is not overstated.
Author Response
RESPONSE LETTER
Thank you for reviewing our manuscript ID IJERPH-1812231. Your comments and suggestions helped us improve this manuscript. Please see below for our point-by-point responses to your comments.
COMMENT #1:
INTRODUCTION is well written. Although the scientific background for the investigation being taken is well explained, I believe the Introduction could benefit from the bringing in the term Fetal programming since this enthity is responsible for the future perspectives of offspring, especially regarding diabetes mellitus, arterial hypertension and obesity. The introduction of fetal programming in Introduction section is justified by the evaluation of Parental history of diabetes (this could encompass gestational diabetes of the mother), Parental smoking at birth (this could encompass Maternal smoking at birth), etc.
Response:
Thank you for the comment and your valuable suggestion. We have now added this point of view in Introduction (Line 82-85).
“Maternal metabolic disturbance, such as GDM, has been known to lead to subsequent fetal metabolic programming, hence, increasing the risk of cardiometabolic disorders in the offspring, including future GDM (transgenerational cycle) [8]”.
- McIntyre, H.D., et al., Gestational diabetes mellitus. Nat. Rev. Dis. Primers 2019;5(1): p. 1-19. doi: 10.1038/s41572-019-0098-8.
COMMENT #2:
MATERIALS AND METHODS are flawlessly written. Population is well defined and sampling approach is clearly written.
Response:
Thank you very much for the comment, we appreciate it.
COMMENT #3:
RESULTS are well presented, without substantional overlapping between data in text and Tables and Figures. However, I believe the addition of p values in Table 1 and Table 2 could increase the quality of presented results.
Response:
Thank you for pointing this out. We have now added p values in Table 1 and Table 2 as suggested and added information under Statistical Analysis regarding this (Line 185-187).
“We used chi-squared tests for categorical variables and t-tests for continuous variables to compare distributions of study participants based on their GDM status (without vs. with GDM).”
COMMENT #4:
DISCUSSION is well balanced, with carefully observed and presented limmitations and therefore it is not overstated.
Response:
Thank you again for your comment and appreciation of this study.

Reviewer 3 Report
This was a retrospective cross-sectional study using the UAE Healthy Future Study (UAEHFS) baseline data (February 2016 to April 2022) on 702 ever-married women aged 18 to 67 years. The authors aimed to estimate the GDM prevalence and examine the associations of maternal early-life risk factors, traditional as well new and so far poorly investigated, such as maternal birthweight, parental smoking at birth, childhood urbanicity, ever-breastfed, parental education attainment, parental history of diabetes, childhood overall health, childhood body size, and childhood height, with later GDM. The study is well designed and has clinical significance. Minor issues: Figures are not clearly visualized, please correct that.
Author Response
RESPONSE LETTER
Thank you for reviewing our manuscript ID IJERPH-1812231. Your comments and suggestions helped us improve this manuscript. Please see below for our point-by-point responses to your comments.
OVERALL COMMENTS
This was a retrospective cross-sectional study using the UAE Healthy Future Study (UAEHFS) baseline data (February 2016 to April 2022) on 702 ever-married women aged 18 to 67 years. The authors aimed to estimate the GDM prevalence and examine the associations of maternal early-life risk factors, traditional as well new and so far poorly investigated, such as maternal birthweight, parental smoking at birth, childhood urbanicity, ever-breastfed, parental education attainment, parental history of diabetes, childhood overall health, childhood body size, and childhood height, with later GDM. The study is well designed and has clinical significance. Minor issues: Figures are not clearly visualized, please correct that.
Response:
Thank you for your constructive comments and suggestion. We have now improved our figures’ visualization (Line 266-267)

Round 2
Reviewer 1 Report
Thank you for your improved manuscript. The research is adequately described and clearly presented.
Author Response
Thank you for reviewing our manuscript ID IJERPH-1812231. Your comments and suggestions helped us improve this manuscript.